assessment tools; culture; developing countries; distress; solastalgia; climate change

**Corresponding author:**
Syed Shabab Wahid;
Email: ssw64@georgetown.edu

# Cultural and contextual adaptation of the Solastalgia subscale of the Environmental Distress Scale in drought-affected Kilifi, Kenya

Syed Shabab Wahid[1] (iD), Linda Norah Khakali[2], Felix Agoi[3], Benjamin Oestericher[4], Emily Mendenhall[4] and Edna N. Bosire[2]

[1]Department of Global Health, School of Health, Georgetown University Medical Center, Georgetown University, Washington, DC, USA; [2]Brain and Mind Institute, The Aga Khan University, Nairobi, Kenya; [3]Department of Population Health, The Aga Khan University, Nairobi, Kenya and [4]School of Foreign Service, Georgetown University, Washington, DC, USA

## Abstract

**Background:** There is an urgent need to measure the psychological toll of climate-related ecological degradation and destruction in low- and middle-income countries. However, availability of locally adapted tools is limited. Our objective was to conduct a transcultural translation and cultural adaptation (TTA) of the Solastalgia subscale of the Environmental Distress Scale (EDS-Solastalgia) in Kilifi, Kenya, which is undergoing transformational changes due to climate change.

**Methods:** We conducted 5 expert interviews, 2 Focus Group Discussions (*n* = 22) and 10 cognitive interviews to solicit feedback on the EDS along the following cultural equivalency domains: Comprehensibility (Semantic equivalence); Relevance (Content equivalence); Response set (Technical equivalence) and Completeness (Semantic, criterion and conceptual equivalence).

**Results:** After an initial translation and back translation of the EDS-Solastalgia, respondents identified several terms that needed to be altered to make the scale understandable, less technical and culturally acceptable. For some items, respondents recommended examples to be included to aid comprehensibility. Feedback from respondents were iteratively integrated into the Swahili EDS-Solastalgia scale, and final endorsement of all changes were confirmed during cognitive interviews.

**Discussion:** The culturally adapted Swahili EDS-Solastalgia scale needs to be tested for its psychometric properties prior to utilization in survey studies to quantitatively establish the burden of climate-related distress and test for associations with common mental health conditions.

## Impact statement

A growing body of evidence indicates that climate change can precipitate novel form of negative affective states, identified in the literature as ecological grief and Solastalgia. These experiences have been described as psychological responses to experiencing or witnessing climate wrought unfettered destruction of nature, species and culturally significant or sacred spaces. Locally and culturally adapted instruments are necessary to quantify this burden and examine its connection to mental disorders, especially in low- and middle-income countries (LMICs), which are facing the harshest impacts of climate change. Few such tools exist in LMICs such as Kenya, or Sub-Saharan Africa in general. In this study, a transcultural translation and cultural adaptation was conducted to adapt the Solastalgia subscale of the Environmental Distress Scale across salient cultural equivalency domains, in Kilifi, a drought-affected region of Kenya. This adapted scale can be utilized in subsequent efforts to examine its psychometric properties and assess the climate-related vulnerability of affected populations. Understanding the local stressors and risk factors is a key priority in global mental health, and can lead to better adaptation of interventions to address novel threats to well-being and mental health imposed by the climate crisis.





## Introduction

Climate change has emerged as a major cause of mental distress worldwide (Hwong et al. 2022). Exposure to increasingly intensifying natural disasters, elevated temperature and indirect effects of climate change, such as forced migration, disruptions to agrarian economies and loss of habitat, etc., represent a constellation of climate-related risk factors that has been associated with poor mental health outcomes (Hayes and Poland 2018; Wahid et al. 2023). Moreover, a growing

body of evidence indicates that climate change can precipitate novel forms of negative affective states, identified in the literature as ecological grief (Cunsolo and Ellis 2018). These states have been described as psychological responses to experiencing or witnessing the unfettered destruction of nature, species and culturally significant or sacred spaces due to climate change. These feelings have been theorized to be closely connected to the construct of '*solastalgia*' or the deep sense of sadness felt in response to witnessing the loss of one's home, traditional ways of life, loss of indigenous knowledge systems and the actual or anticipated loss of solace that one previously found in home environments (Albrecht et al. 2007). Solastalgia can be understood in reference to nostalgia, where the latter is the emotion of missing the home environment one lived in previously, solastalgia is experienced by those who still reside in their home locations, but experience negative emotions due to witnessing the gradual degradation, degeneration and destruction of one's home environment. Solastalgia has come under increased focus since being first defined by Albrecht, and a diversity of literature around the construct has steadily accrued from across the world. In a scoping review of this extant literature on solastalgia, Galway et al. (2019) highlight the critical role it has in fully understanding the environment-health-place nexus in the context of worsening climate change. However, they emphasize the need for additional research of the construct across a greater diversity of peoples and places (Galway et al. 2019), that would help refine its theoretical and conceptual premise and assumptions. Moreover, as emerging research has raised the possibility of solastalgia to be connected to deeper states of mental distress, including potentially clinically salient outcomes, such as depression, suicide and substance abuse, further examination of this connection is warranted as well (Barton 2017).

To examine solastalgia and other environment-related cluster of emotional states, Higginbotham et al. (2006) developed the Environmental Distress Scale (EDS). Informed via ethnographic work in the Upper Hunter Valley region of Australia, the EDS functions as a novel index capturing the psychological toll of disruptions to ecological systems. The instrument contains a subscale on solastalgia (EDS-Solastalgia) in addition to items capturing participants' perception of hazards, appraisal of threats, felt impact of negative environmental changes and environmental action. The EDS has subsequently been identified and utilized as a candidate scale to capture distress connected to other environmental disasters and disruptions (Warsini et al. 2014; Eisenman et al. 2015).

While efforts to examine solastalgia have been pivotal in illuminating these novel forms of climate-related psychological states, the vast majority of research on the topic has been conducted in higher income countries with individuals relatively buffered from the direct impacts of climate change, or within minority and indigenous populations in such countries (Benham and Hoerst 2024). The negative impacts of climate change are disproportionately worse, and are predicted to worsen substantially, in low- and middle-income countries (LMICs), which are ill-equipped with resources to deal with the climate crisis (Sharpe and Davison 2021). There is little to no evidence on how climate change precipitates such novel forms of distress in LMIC. Therefore, there is an urgent need to measure the psychological toll of ecological degradation and destruction in such countries which constitute the frontlines of climate change. A measurement tool such as the EDS-Solastalgia can be a good candidate instrument to adapt and quantify the magnitude of such climate-related distress within disaffected LMIC population.

Adapting an existing tool has been cited as preferable to developing a wholly new instrument as it reduces complexity of tool development, saves time and costs, while allowing for comparisons of results with other regions and populations (Epstein et al. 2015; Nyongesa et al. 2022). Due to considerable differences in culture and context from the original settings of the development of the EDS-Solastalgia, there is a need for transcultural translation and adaptation (TTA) to ensure cultural equivalency of the instrument before its use in LMIC settings (van Ommeren et al. 1999). This involves taking into consideration subjective experiences of emotions, cognitions and even somatic aspects of mental distress and disorder, which are considerably shaped through the local cultural perceptions surrounding mind and body functions and disruptions (White 1992). Moreover, linguistic features of local terminology used in expressing distress in global settings, including substantial heterogeneity in the range of experiences that can be subsumed under such terms, are heavily shaped by culture as well (Mendenhall et al. 2019; Wahid et al. 2021). Accordingly, tools that were developed in one cultural setting may fail to fully and accurately capture and measure mental health issues in other cultural settings if these are not carefully and systematically adapted incorporating local cultural aspects (Ali et al. 2016).

TTA procedures of instruments involve adaptation along several key dimensions to ensure cultural equivalency (de Lima Barroso et al. 2018). These include cultural acceptability, comprehensibility, relevance, completeness and considerations of response options for scale items (Flaherty et al. 1988; van Ommeren et al. 1999). Preparation of tools along these dimensions necessitates involvement of multi-stakeholder perspectives. These include populations who hold direct lived experience of the psychosocial domains under investigation to consider the needs of such specific groups and ensure locally suitability and acceptability. Opinions from mental health experts are also recommended as they can provide input on both cultural equivalencies, as well as on the preservation of clinical and psychometric properties of the translated tool as originally intended by tool developers (Kaiser et al. 2013; Repo and Rosqvist 2016).

As the EDS-Solastalgia includes questions that inquire about lived experience of emotions and cognitions contextualized within the surrounding physical and social environments, it requires careful consideration of culture and adaptation before administration in survey methods. Such procedures have been utilized previously for the EDS. For example, following a volcanic disruption in Indonesia (Warsini et al. 2014), researchers utilized a systematic adaptation following transcultural adaptation procedures to prepare the tool and measure the psychological impact of the disaster. Few studies have culturally adapted tools for understanding environmental distress in rural regions in Kenya, or in sub-Saharan Africa more broadly. This prevents a more robust understanding of how people perceive and experience climate change how the effects of such experiences can be measured, and ultimately, considerations for the development of culturally acceptable interventions to mitigate psychological impacts. In the current study, we utilize standard procedures of TTA (Flaherty et al. 1988; van Ommeren et al. 1999) to adapt the EDS-Solastalgia to the context of rural Kenya. We focus on the rural coastal county of Kilifi located in Southeastern Kenya, a semiarid climate-affected region that is undergoing transformational changes due to a prolonged drought that has disrupted local ecologies, economies, traditional cultural roles and ways of life, in profoundly negative ways. As droughts negatively transform the environment over time, as opposed to acute climate disasters such as hurricanes or heat waves, the solastalgia subscale of the EDS, as opposed to the other sub-constructs of hazard, threats and so forth, could potentially be a salient measure in Kilifi to capture the psychological impacts of the slow-paced

degradation of the place-environment-mental health nexus. In partnership with the Aga Khan University (AKU) in Nairobi, and the Kaloleni/Rabai Community Health and Demographic Surveillance System in Kilifi County, a cultural adaptation of the EDS-Solastalgia was conducted in Kilifi. Our objective was to conduct focus group discussions (FGDs), cognitive interviews (CIs) and expert interviews, to iteratively develop an improved understanding of the culturally adapted items of the EDS-Solastalgia and associated meanings. This resulted in a revision of the EDS-Solastalgia to a locally acceptable and comprehensible Swahili version.

## Materials and methods

This study utilized globally established and widely used standards of TTA of instruments as established by Flaherty et al. (1988) and van Ommeren et al. (1999), and as outlined in the COnsensus-based Standards for the selection of health status Measurement Instruments guidelines (Mokkink et al. 2010). Using a qualitative approach, we considered perspectives from mental health experts and community members on the EDS-Solastalgia scale items along the following cultural equivalency domains of the TTA procedure: Comprehensibility (Semantic equivalence); Relevance (Content equivalence); Response set (Technical equivalence); Completeness (Semantic, criterion and conceptual equivalence) of the EDS-Solastalgia scale.

### Setting and study site

This study was conducted across two subcounties, Rabai and Kaloleni, in Kilifi County.

More than half of the community depends on low-input rain-fed agriculture in Kilifi County for most of their household income (Ngugi et al. 2020). The majority of inhabitants of Kilifi County belong to one of Kenya's oldest communities, the Mijikenda peoples. They have a centuries-long history of horticulture and pastoralism, trade relationships with the Swahili coast, and economic, political and military alliances (Keida 2022). They have also endured threats to their way of life, from forced displacement from the northern Somali coast, to enslavement on large-scale plantations, to British colonizers that dispossessed them from their lands (Keida 2022). They now face climate changes that have made their lands increasingly inhospitable to their way of life (Chemuku et al. 2021).

The solastalgia subscale as a measure of the psychological impact of climate change is especially salient for the Mijikenda people as Kilifi contains historic forests and locations, which have earned UNESCO status as biocultural and world heritage sites (Chemuku et al. 2021). The Kaya forests in Kilifi are deeply entrenched in traditional Mijikenda culture. The traditional cultural concept of "*Mudzini,*" comprises Mijikenda elders' worldview and understanding of well-being grounded in human–nature harmony and coexistence. This perspective informs Mijikenda interactions within the landscape as shaped by cultural values of *kufaana* (reciprocity) and *soyosoyo* (equilibrium) between humans and nature, *umwenga* (solidarity) among individuals with shared interests and *kushirikiana* (collectiveness) within the community (Chemuku et al. 2021). Together, these principles have promoted deep endorsement by the Mijikenda toward sustainable resource management, social cohesion and the preservation of traditional knowledge and practices (Chemuku et al. 2021). Given the historic connection that the Mijikenda have with the land, forests, animals, rivers and so forth, it can be hypothesized that degradation of these culturally sacred entities due to ongoing drought and other climate stressors is likely to have profound psychological consequences.

The study was conducted in parallel with the existing Kaloleni/Rabai Community Health and Demographic Surveillance System (Ngugi et al. 2020). This cohort is situated within Kilifi county on the coast of Kenya, and was created to capture information using the governments' Community Health Strategy (CHS) by the AKU and its partners at the Kaloleni and Rabai Sub-County Health Management Offices. The Kenyan government's national strategic response to poor health indicators (defined by a decline in population health from the 1990s) can be understood in the CHS, which was designed to develop the capacity to deliver basic health services through community health volunteers (CHVs) at the community health facility interface. The CHVs are assigned to and serve designated community health units, which are clusters on average of 1,000 households and 5,000 people within a geographically defined area, which is aligned to an administrative sub-location. This surveillance system brings together longitudinal information about individuals using unique identifiers, and then follows up with residents of 112 villages in 10 community units (Ngugi et al. 2020).

### Participants, data collection and data analysis

We used a purposive sampling strategy to identify and recruit participants in this study (Patton 2014). Adult individuals who were long-term residents of the two subcounties (Kaloleni and Rabai) were eligible for recruitment, as they may have experienced changes to the surrounding environment due to climate change over a few decades, and could reflect in interviews how such changes have negatively transformed the land and the local ways of life, and any associated psychological impacts. Participants were recruited in equal numbers from the two subcounties to reflect rural and peri-urban contexts. Participant recruitment for the Kilifi respondents was done by four ($n = 4$) trained CHVs, two from each subcounty, who have long-standing relationships with the community.

Prior to initiation of the data collection, informed consent was obtained, followed by collection of basic demographic information. For the CIs, a monitoring form was used to capture any changes, suggestions or recommendations. Given the objective and topic of discussions, several respondents informed the data collection team about discomfort in having a recording device, and accordingly, FDSs and CIs were not recorded. Instead the research team made detailed observations in field notes capturing the insights shared by respondents during the process of the tool demonstration and discussion along the key cultural equivalency domains outlined in Table 1: Comprehensibility (Semantic equivalence); Relevance (Content equivalence); Response set (Technical equivalence) and Completeness (Semantic, criterion and conceptual equivalence) of the scale (Flaherty et al. 1988; van Ommeren et al. 1999).

The TTA process followed the steps outlined in Table 1. In Stage-1, to prepare the scale, one local bilingual researcher conducted a translation of the EDS from English to Swahili, a commonly used national language in Kenya. The researcher then back-translated the Swahili EDS-Solastalgia scale to English while identifying differences in words from the original English version. The translation and back-translation were checked and verified by a team of bilingual Kenyan researchers, including several coauthors of the current study (EB, FA and LNK). Afterward, in Stage-2, we conducted five expert interviews, including a clinical psychologist, two nurses and two public health officers. These participants were recruited on the basis that they had worked in Kaloleni

**Table 1.** Stages of transcultural translation and cultural adaptation (TTA) of the Solastalgia-EDS in Kilifi, Kenya, and respondents who participated in each stage, that is, experts, focus group discussion participants and cognitive interview respondents

| TTA stages | Kaloleni subcounty | Rabai subcounty | Participant characteristics | Total (*n*) |
|---|---|---|---|---|
| TTA Stage–1: Translation and back-translation | N/A | N/A | One researcher conducted translation and back-translation which was reviewed by a team of three bilingual researchers | 4 |
| TTA Stage–2: Expert interviews | 2 | 3 | One psychologist, two nurses and two public health officers; all experts have worked in Kaloleni and Rabai subcounties for greater than 2 years | 5 |
| TTA Stage–3: Focus group discussions | 1 (female, *n* = 10) | 1 (male, *n* = 12) | Men and women aged 18 years and above, who have lived in these settings for more than 10 years | 22 |
| TTA Stage–4: Cognitive interviews | 5 (female, *n* = 3; male, *n* = 2) | 5 (female, *n* = 2; male, *n* = 3) | Men and women aged 18 years and above, who have lived in these settings for more than 10 years | 10 |

and Rabai subcounties for 2 years or longer, and had previously been involved in projects involving mental health research in these contexts. The experts reviewed each of the translated items in the EDS-Solastalgia scale assessing the cultural equivalency domains referenced above. Participants' insights were key in reviewing and assessing the Swahili and English EDS-Solastalgia scale and providing comments suggesting any new changes.

In Stage-3, we conducted two FDSs (FGDs, *n* = 22), one with women and another with men, in Kaloleni and Rabai subcounties, respectively. FGD discussions took approximately 120 min to complete. One FGD was conducted in a community setting and another in a nearby local dispensary. Discussants were queried about each scale item of the EDS-Solastalgia, to elicit impressions on the cultural equivalency domains for the terminology of the translated version of the scale. Discussants were given time to provide any recommended changes including identifying technical words, words that were difficult to understand, or words that meant something different in their local contexts than as intended for scale purposes. Responses were summarized and preserved in field notes by the research team.

Finally, in Stage-4, the team recruited 10 participants, who were long-term residents of Kilifi (8+ years) to take part in CIs assessing both the latest version of the Swahili and English EDS-Solastalgia scale. CIs focused on respondents' understanding of the specific wording of the tools. Each item was spoken out loud and presented in printed form for respondents to review and to gauge their comprehension and identify any problematic words. Respondents rephrased the items in their own words for each EDS-Solastalgia item. Respondents provided ratings of items using the response set of the scale, and indicated reasons behind their ratings, and what experiences would be necessary for them to consider either lowering or increasing their ratings. Any final changes, recommendations or suggestions were recorded.

Data from expert reviews, FGDs and CIs were iteratively analyzed using a deductive codebook comprising the cultural equivalency domains outlined above. We monitored the translations during data collection using the translation monitoring form developed by van Ommeren et al. (1999). Response set items were not identified as problematic by CI participants. Data were extracted during each round of collection from each activity, and deductively charted into the cultural domains in a spreadsheet, which was managed via Microsoft Excel. Given the formulaic aspects of feedback on tool items, and as data included recommendations, which broadly aligned across the various data collection modalities and respondents, all recommendations were retained, and conventional thematic analysis was not deemed to be necessary. All changes made to working version of the translation included synthesized findings from the previous TTA stages. The endorsement of the final translation by CI participants without any major changes was indicative of consensus across the cultural equivalency domains.

## Results

The sample characteristics of the study are presented in Table 2. We strived to attain equal representation from both subcounties, from experts and residents alike, and toward gender parity in the sample. Around 22 community members participated in the focus groups, and 10 were recruited for the CIs. Four experts with medical backgrounds and experience in conducting mental health research in the subcounties were recruited for the expert review. Detailed information of all changes incorporated in the TTA stages can be found in Supplementary Annex-1.

### *TTA Stage-1: Translations*

The translated Swahili EDS-Solastalgia scale was back translated to English to ensure that the meanings did not change/deviate from the original intended meaning. In the back-translation phase, a few words were changed from the original version. For example, 'unwelcome changes' were replaced with 'unacceptable changes.' It was perceived that 'unwelcome' was not a culturally salient word, but 'unaccepted' was commonly used and could be easily translated to Swahili without altering the original meaning. Similarly, 'aspects of this area that I value are being lost' were replaced with 'things that I value/treasure in this area are disappearing.'

### *TTA Stage-2: Findings from expert interviews*

Interviews with mental health experts indicated that some words in the EDS-Solastalgia were too technical, not comprehensible, and sometimes not culturally relevant, as summarized in Supplementary Annex-1. In addition, when translated into Swahili and then back translated to English, some words lost the original English meaning. For example, the word 'undermine,' in the first statement, when translated to Swahili was 'kudhoofika.' However, experts felt that the word 'kudhoofika' was too technical and most people would not easily understand it. 'Kudhoofika' is used mostly in attribution to suffering as experienced by animals or humans. Hence, one expert recommended replacing the word 'kudhoofika' with 'dhalilishwa,' which is closer to 'undermine' in meaning. The word 'aspects' in the second and third EDS-Solastlagia items was initially translated to 'vipengele' in Swahili. However, two experts felt that the word 'vipengele' was not locally appropriate syed could be difficult to understand. In addition, when back translated to English, 'vipengele' could mean 'indicators' which distorts the original English meaning

**Table 2.** Sample characteristics

| Sample sociodemographic characteristics | Expert interviews (*n* = 5) | FGD participants (*n* = 22) | Cognitive interviews (*n* = 10) |
|---|---|---|---|
| **Age** | | | |
| 18–35 | 20% (1) | 18% (4) | 20% (2) |
| 36–45 | 60% (3) | 36% (8) | 50% (5) |
| 46–55 | 20% (1) | 27% (6) | 20% (2) |
| 56+ | 0 | 18% (4) | 10% (1) |
| **Gender** | | | |
| Men | 40% (2) | 54% (12) | 50% (5) |
| Women | 60% (3) | 46% (10) | 50% (5) |
| **Education** | | | |
| Primary school | 0 | 18% (4) | 20% (2) |
| Secondary school | 0 | 41% (9) | 60% (6) |
| College/university | 100 (5) | 31% (7) | 20% (2) |
| **Employment/labor** | | | |
| Unemployed | 0 | 23% (5) | 20% (2) |
| Self-employed/farmer | 20% (1) | 41% (9) | 50% (5) |
| Casual laborer | 0 | 36% (8) | 30% (3) |
| Permanent with pension | 80% (4) | 0 | 0 |
| **Residency** | | | |
| Rabai | 60% (3) | 54% (12) | 50% (5) |
| Kaloleni | 40% (2) | 46% (10) | 50% (5) |
| **Years lived/worked in Kaloleni/Rabai** | | | |
| 1–4 years | 20% (1) | 9% (2) | 0 |
| 5–9 years | 60% (3) | 23% (5) | 20% (2) |
| Over 10 years | 20% (1) | 68% (15) | 80% (8) |

of the word 'aspects.' Thus, it was recommended that the word be replaced with a simpler word such as 'vitu,' which when back translated to English would mean 'things.' The word 'upset' in the third statement was initially translated to 'kukasirishwa' in Swahili. Experts felt that the word 'kukasirishwa' was not a good fit for the local context and specifically in relation to climate change events because this word was linked to getting irritated when someone/somebody wrongs another person. Experts argued that would not necessarily be the case for climate change events. Hence, experts recommend using Swahili word 'sijafurahishwa' which would translate into English as 'unhappy,' which better retained the original meaning/intention of the item.

### TTA Stage-3: Findings from FDSs

Similar to what we gathered from expert interviews, discussants in the two FGDs recommended various changes in the EDS-Solastalgia scale. Some words were found to be technical, not culturally relevant, or could not be easily comprehended in the local context. For example, discussants in the two FGDs raised similar issues or perceptions on words such as 'undermined' which in Swahili would translate to 'kudhoofika'. They recommended replacing 'kudhoofika'

with a simpler word. In addition, the phrase 'unwelcome changes' (Swahili: 'yasiyokubalika') was identified as unsuitable in regards to climate-related events. FGD discussants argued that people did not have power to influence things that would be considered welcome or unwelcome. In addition, the phrase 'unwelcome changes' was found not relevant in the context of weather or environmental change. Hence, discussants recommended using the phrase 'yasiyo ya kawaida' which translates to 'unusual changes' in English, as a suitable alternative translation. In the second EDS-Solastalgia item, discussants in both groups recommended replacing the word 'vipengele,' which in English means 'aspects,' with a simple word such as 'vitu', which translated in English to mean 'things.' This recommendation aligned with what had been recommended by experts previously. In addition, they recommended replacing the word 'vinatoweka,' which in English means 'disappearing,' with 'vimeisha.' Discussants argued that 'kutoweka,' the root word from which 'vinatoweka' is derived, would loosely be used locally to signify a living thing, for example, human beings or animals running away or disappearing. 'Kutoweka' also has the implication that the thing that has disappeared can be found. However, the word 'kuisha' can be used in reference to both flora and fauna, in addition to geographical features, that may completely become extinct/destroyed as a result of harsh climatic changes. Thus, discussants recommended using the word 'vimeisha' to replace 'vinatoweka' to better convey that meaning. Also, it was recommended that there was a need to add examples on the things that were disappearing to make the statement more relevant, complete and comprehensible.

For EDS-Solastalgia item-3, discussants in both FGDs recommended replacing 'vipengele' with a simple word such as 'vitu' as discussed above, and provide examples to specify what aspects are being talked about for cultural relevance. For EDS-Solastalgia item-4, discussants argued that Swahili word 'mahali hapa' which in English means 'this place' was restricted to only a very small area, such as a room in a house. They recommended replacing 'mahali hapa' with 'eneo hili' which would communicate a larger area or neighborhood or geographical location. They also recommended replacing Swahili word 'awali,' which in English translates to 'before', with 'hapo mwanzo' which in English would mean 'in the beginning.' While 'awali' and 'Hapo mwanzo' are synonyms, the latter was said to be used more commonly in the study settings.

For EDS-Solastalgia item-5, participants from both FGDs recommended use of a more polite word to replace 'upset.' Respondents felt that the word 'upset' was too harsh, especially in relation to climate change events, and would conventionally be used in relation to someone not being happy due to interpersonal rude behavior, disagreements and so forth. They suggested replacing 'upset,' which in Swahili means 'nimekasirishwa,' with 'unhappy' which in Swahili would translate to 'sijafurahishwa'. For item-6, discussants recommended replacing the Swahili word 'mtindo,' which translates to English as 'my lifestyle,' with 'hali.' Discussants argued that 'mtindo' was mostly used to indicate facets of one's individual lifestyle such as fashion, dressing style, food habits and walking styles, which would not be appropriate in consideration to climate change events. Similar to item-4, discussants also suggested replacing Swahili word 'eneo langu' with 'eneo hili'. The word 'eneo' in English means 'place'; however, 'eneo langu' limits the participant to their immediate personal space, such as one's household, while 'eneo hili' can be used to refer to a geographical location, for example, a neighborhood or local area that is shared by a community.

For EDS-Solastalgia item-7, similar to earlier items, FGD 2 discussants recommended replacing Swahili word 'vipengele' with 'vitu' to mean 'aspects.' Respondents recommended changing the

word 'asili' which translates to mean traditional things (traditional clothes, cooking utensils etc.) to 'maandhari' which is closer in meaning to 'nature/environment.' For item-8, FGD 2 recommended replacing Swahili word 'huzuni,' which in English translates to 'sad,' with 'kusikitika' which in English would still mean 'saddened' but would be more appropriate for the EDS-Solastalgia. They argued that the word 'huzuni' was conventionally used in reference to something severe such as death, while the word 'kusikitika' is more general, and can be used to include loss of natural environment, and also, would encompass some aspects of empathy. For item-9, FGD 1 recommended replacing Swahili word 'yasiyokubalika,' which in English translates to 'unacceptable,' with 'yasiyo ya kawaida' which in English would translate to 'unusual.' 'Unacceptable' was found to be a poorer fit in the context of climate change events, given that people did not have power to control these events.

### TTA Stage-4: Findings from CIs

After making all changes as suggested by experts and FGD discussants, the researchers including two local leads consolidated all changes and came into consensus on what to implement and what not to implement. In many cases, recommendations provided across the different groups were similar. On a few occasions where there were discrepancies, the researchers sat down with two local experts in the community to deliberate on what to keep, to a point where consensus was reached prior to conducting the CIs. Thereafter, the newly updated scale (English and Swahili versions) incorporating changes from expert review and FGD feedback was administered to participants in CIs. All participants who took part in CIs found each statement of the tool to be comprehensible, acceptable, cultural relevant and easy to formulate responses to, given the original Likert-style response set options. When asked to provide reasonings for their score, they indicated it to reflect their own experience, and mentioned the situation would have to improve or worsen for their assigned scores to increase or decrease accordingly. They confirmed their agreement with the changes that had been suggested by FGD participants and expert reviewers. The final translated Swahili version of the EDS-Solastalgia is presented in Table 3.

### Discussion

We conducted a TTA of the EDS-Solastalgia scale in Kilifi Kenya using expert review, FGDs and CIs. We identified a number of words that were either highly technical, not relevant or culturally incongruent with local experiences of respondents of Kilifi. Some items were translated with accuracy but were not found to be relevant considering local spoken language considerations, and had to be simplified with locally relevant terms for comprehensibility. Some items were found to be incomplete without the addition of examples that are locally relevant, after which FGD respondents and CI participants indicated the items to be comprehensible and complete.

Climate change and related distress is an important developing area. Our research indicates that stressors and impacts of 'climate change' need to be grounded and contextualized in locally salient experiences to clearly communicate what changes are manifesting in local circumstances for local respondents to connect with their lived experience and explore related psychological impacts. While examples of rivers or coconut trees were necessary to make the scale

**Table 3.** Transculturally translated and adapted Solastalgia subscale of the Environmental Distress Scale (EDS-Solastalgia) in Swahili

| Original EDS-Solastalgia items in English | Final version in Swahili | Back-translation into English of final Swahili version |
|---|---|---|
| My sense of belonging to this place has been undermined by unwelcome change | Hisia yangu ya kuwa mwenyeji wa eneo hili imedhalilishwa na mabadiliko yasio ya kawaida. | My sense of belonging to this place has been undermined by unusual changes |
| I am sad that familiar aspects of this place are disappearing (e.g., animals, plants, landmarks and open space) | Nina huzuni kwamba vitu vinavyojulikana vya mahali hapa vinaisha/vinapotea (k.m., wanyama, mimea, alama za jiographia, nafasi za wazi). | I am sad that familiar things in this place are getting depleted/lost (e.g., animals, plants, landmarks and open spaces] |
| I am worried that aspects of this area that I value are being lost | Nina wasiwasi kuwa vitu vya eneo hili ambavyo navithamini (km mito,mifugo, misitu,minazi) vimeisha/vimepotea | I am worried that the things that I value in this place (e.g., rivers, animals, forests and coconut trees) are being lost/depleted |
| I miss having the peaceful feeling that I once enjoyed by being in this place | Ninakosa kuwa na hisia ya amani/utulivu ambayo hapo mwanzo nilifurahia kwa kuwa mwenyeji wa eneo hili. | I miss having the peaceful feeling that I enjoyed in the beginning by being a resident of this place |
| I am upset at the way this area looks now | Sijafurahishwa na jinsi eneo hili linavyoonekana kwa sasa. | I am unhappy with the way this place looks now |
| My lifestyle is being threatened by change in my local area | Hali yangu ya maisha inatishiwa na mabadiliko katika eneo hili. | My state of life is being threatened by changes in this place |
| Unique aspects of nature that made this place special are being lost forever | Vitu vya kipekee vya maandhari vilivyofanya mahali hapa kuwa maalum vinapotea. | Unique aspects of nature that made this place to be unique are being lost |
| I am saddened by unwelcomed change I see in my landscape | Nimesikitishwa na mabadiliko yasiyo ya kawaida ninayoyaona katika mazingira yangu. | I am saddened by unusual changes that I can see in my environment |
| I feel powerless to stop unwanted changes to this place | Ninahisi kutokuwa na uwezo wa kukomesha mabadiliko yasiyo ya kawaida mahali hapa. | I feel powerless to stop unusual changes to this place |

relevant to Kilifi populations, the use of the EDS-Solastalgia in other contexts would necessitate incorporating examples of environmental impacts in that area, for the scale to be relevant in measuring the impacts of climate change in that area. For example, disappearance of coconut trees may not be relevant in Nairobi city, or in the northern frontier of Canada and so forth, and culturally equivalent replacements must be used.

In previous efforts of mental health tool adaptation, CIs were found to add substantial value in the translation and adaptation process (Nyongesa et al. 2022). However, in our study, the majority of recommended changes emerged from expert reviews and from

FDSs. Expert reviews have been identified to offer significant value in such procedures (Epstein et al. 2015), which we confirm in the current study. Previous research in tool adaptation has raised concerns about the limitations of FGD formats as potentially stifling the opinion sharing by some participants due to group dynamics (Nyongesa et al. 2022). CIs that use a one-on-one format were identified as revealing additional insight perhaps due to the personal and anonymous nature of the format. However, in the current study, our experience found FGD respondents in Kilifi to be vocal about the changes they identified to be necessary to make the EDS-Solastalgia locally and culturally salient. The universal confirmation by CI participants of the scale as comprehensible, relevant, acceptable, easy to respond to and complete, acts as a strong endorsement of the changes introduced based on the expert and FGD feedback.

As Kenya continues to face unprecedented climactic changes in its environments, disasters and disruptions to ways of life, affected populations will continue to experience psychological impacts. There is a need to understand the depth of these experiences in its totality, and examine the connection of climate wrought psychological issues with clinically salient outcomes such as depression, anxiety, trauma, suicidality, aggression, violence and substance abuse. This culturally adapted EDS-Solastalgia scale will now be tested for its psychometric properties in a forthcoming community-based quantitative study. If it is found to be psychometrically valid, it can be incorporated in future survey studies to examine its relationship with common mental disorders and other risk factors. The use of an EDS-Solastalgia scale can also identify specific subgroups who may be more vulnerable to experiencing such forms of distress, and inform and prepare health systems for necessary service provision. While mental health service delivery is a core component of climate adaptation efforts, the problem ultimately transcends the health system, and will require efforts using a cross-sectoral approach, including social services, financial and social protection initiatives, education, agriculture and so forth, mobilized across all social ecological levels, starting from the individual, communities, neighborhoods, institutions and policy. Utilizing the EDS-Solastalgia to quantify the burden of climate-related distress can be instrumental in informing such approaches.

## Strengths and limitations

The current study utilized expert reviews, FGDs and CIs to culturally adapt the EDS-Solastalgia scale for rural Kilifi, Kenya. The multi-stakeholder perspectives that informed this process is a key strength, as it allows for triangulation of the changes made to the instrument in a progressive way, with earlier changes being iteratively incorporated and reviewed in subsequent efforts. Limitations include the refusal of participants to record the FGDs and CIs, which was partially mitigated by having one facilitator conduct the interviews and a separate researcher taking notes using the standardized translation monitoring form (van Ommeren et al. 1999) which has been widely used in TTA of psychological instruments worldwide. An additional limitation is the gender composition of the sample, with a male-only group in Rabai, and a female-only group from Kalolni, participating in FGDs. Kilifi itself is a small county of Kenya, and the two subcounties are very similar considering the gender, sociocultural and normative contexts. Additionally, there were no contradictions identified in the recommendations from the two FGDs. Accordingly, we consider the risk of gender-related biases to have been minimal.

## Conclusions

The TTA procedures utilizing FGDs, CIs and expert review illuminated a wide range of terminology and cultural aspects that needed to be incorporated to make the EDS-Solastalgia culturally salient for Kilifi, Kenya. This adapted scale now needs to be assessed for validity of its psychometric properties before it can be utilized to quantitatively establish the burden of psychological ill-being attributable to novel forms of climate-related distress, namely solastalgia. Taken together, these efforts can yield meaningful insights into the experiences of distress for residents of Kilifi, Kenya, who are facing worsening realities on the frontlines of the climate crisis.

**Open peer review.** To view the open peer review materials for this article, please visit http://doi.org/10.1017/gmh.2025.8.

**Supplementary material.** The supplementary material for this article can be found at http://doi.org/10.1017/gmh.2025.8.

**Data availability statement.** The data that support the findings of this study are not publicly available due to restrictions related to privacy and ethical considerations. Participants provided informed consent on the basis that their responses would remain confidential and would not be shared outside the research team. As a result, the raw data cannot be disclosed. For further information about the study, interested parties can contact the corresponding author.

**Author contribution.** Conceptualization: SSW and EM. Methodology: SSW. Data collection: ENB, LNK, FA and BO. Data Analysis: ENB, LNK and FA. Writing original draft: SSW, ENB, LNK and FA. Writing – review and editing: All authors. Project administration: EM and ENB.

**Financial support.** Not applicable.

**Competing interest.** The authors declare no conflict of interest.

**Ethics statement.** The study adhered to the principles of ethical research conduct as established in the Declaration of Helsinki for medical research involving human subjects. Ethical approval for this study was provided by the institutional review boards of Georgetown University, USA (STUDY00006498) and The Aga Khan University, Kenya (2023/ISERC-32 (v2)).

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
