## [Reviewer Report]

The manuscript titled Cultural and contextual adaptation of the Environmental Distress Scale in drought-affected Kilifi, Kenya: A transcultural translation and cultural adaptation, is a process of translation and cultural adaptation of a scale for the evaluation of aspects related to environmental processes. The authors present a manuscript whose theme is current and important in the environmental and geopolitical context. The objective technical language is Clara discussing the aspects that justify the need for the use of a scale to assess environmental effects by residents of communities in Kenya. They use pertinent and congruent references with the theme, most of which are more than 5 years old, which weakens the exposed foundation.

The present review seeks to contribute to the qualification of the manuscript and the possibility of disseminating its use to the international community in view of the importance of the theme.

It is suggested that the authors rethink the issue of the title so that there is no circularity of terms, the reviewer takes the liberty of presenting this suggestion Cultural adaptation of the Environmental Suffering Scale.

Likewise, there are unclear expressions that can be objectified (translation and rigorous cross-cultural adaptation), the rigorous expression does not proceed to what is suggested to be suppressed in the entire manuscript.

In the impact statement, line 43 reads “In this study, a rigorous cross-cultural translation and cultural adaptation was performed to adapt the Environmental Distress Scale....”(it is suggested In this study, cultural translation and adaptation was performed for the Environmental Distress Scale...")

In the introduction, line 19, where it reads “... The procedures of cross-cultural translation and cultural adaptation of...” consider revising to “... The procedures of translation and cultural adaptation...”

In line 25 participants, the authors characterize both the translation and the back translation, being performed by a researcher, which the international literature characterizes as requiring more than one translator to verify the adequacy of the translation and back translation process. The proofreader understands the difficulty, but given the conditions, it is possible for another translator to participate in the process and thus characterize different nuances of the translation process.

The other methodological aspects, such as interviews with experts, the performance of focus groups, in-depth interviews, are adequate and congruent with what is recommended by the international literature. As for the cognitive interviews, the participation of only 2 community members was verified, and the results show a discrepancy in the presentation of results compared to the previous ones with experts and focus group. This is the most important aspect, which is the understanding by the target population, and, in this sense, the perceptions, difficulties and misunderstandings could be better described in order to characterize the perception and understanding of the members of the communities.

The tables referring to the composition of the participants are presented (where the reviewer cannot observe psychosocial characteristics - education, age and work activities that would characterize the representativeness of the participants to the reader). Table 2 is very extensive and compromises the reading follow-up. It is suggested that the authors be more objective in the exposition. Table 3 presents the items of the instrument in its English and Swahili versions. All of them are important, but the idea of more objectivity is reinforced in table 2.

The reviewer notices the absence of the presentation of the results of the scale in quantitative terms, since these in the conclusion are proclaimed as possible to be verified for the measurement of malaise. It would be interesting to use a pilot administration to verify the consistency of the responses and the resulting metrics.

In the conclusions, the authors present very succinctly the characteristics of the scale and the possibility of its use, however the reviewer emphasizes the possibility of expanding the findings in the conclusion, while suggesting restraint in the use of the scale to assess the discomfort related to environmental aspects. This is due to the absence of aspects of criterion validity, which was not the object of the present study.

---

## [Reviewer Report]

Review report on ”Cultural and contextual adaptation of the Environmental Distress Scale in drought-affected Kilifi, Kenya: A transcultural translation and cultural adaptation” (GMH-2024-0107) submitted to Cambridge Prisms: Global Mental Health.

The authors state in the abstract that their objective was “to conduct rigorous transcultural translation and adaptation (TTA) of the Environmental Distress Scale (EDS) in Kilifi, Kenya” and concludes that “The culturally adapted Swahili EDS scale can be utilized in survey studies to quantitatively establish the burden of climate-related distress and test for associations with common mental health conditions.” However, this is not possible considering the weaknesses of the research presented in this study.

There are several problems with this study, such as the cognitive interviews not being recorded (page 7) resulting in possible biases in the interpretations, and not being possible to return to the interviews and listen to these again in case one has missed something. Moreover, they recruited participants from two sub-counties (Kaloleni and Rabai), including both men and women. However, in Kaloleni only women were recruited, and in Rabai only men, resulting in skewed groups implying a risk of bias.

However, the main weakness is that the authors have not psychometrically tested the culturally adapted Swahili EDS scale. While a transcultural translation and adaptation is necessary as a first step, the most important step is the psychometrical testing of the translated and adapted scale, which this study is lacking. The Swahili EDS scale can thus not be utilized in survey studies to quantitatively establish the burden of climate-related distress and test for associations with common mental health conditions, contrary to what the authors claim. In the current state, the paper is thus not acceptable for publication.

Specific comments:

The paper needs a thorough proof-reading to correct grammatical errors and improve the readability of the text, which is sometimes hard to understand.

Page 6: The entire first paragraph of the “Setting and study site” section lacks references, please add references for all statements made in this paragraph.

Page 8: On this page it is stated that “Around 22 community members participated in the focus groups” but on page 7 it was stated that the authors “conducted two focus group discussions (FGDs, n=20)”. Which number is correct?

Page 12: There should be references for the sentence “In previous efforts of mental health tool adaptation, cognitive interviews were found to add substantial value in the translation and adaptation process.”

---

## [Reviewer Report]

1. The original EDS instrument is consist of 8 domain (place attachment, frequency and threat, impact and so on) with total items around 117 questions. Why did the authors conduct cultural and contextual adaptation only on Feelings about changes in Kenya due to climate change?

2. I think something missing here is the process of adaptation it self. Whose instrument adaptation standards were used in this research?

3. Did the authors conduct scoping and observation phase about environmental degradation/changes happened in Kenya due to climate change?

---

## [Reviewer Report]

Cultural and contextual adaptation of the EDS in a drought-affected Kilifi, Kenya: A transcultural adaptation

General Comments

The value of this qualitative study is that it applies the rigorous translation methods of cross-cultural psychology to begin the design of a new psychometric scale measuring environmental distress arising from climate change degradation in an African culture.

The study demonstrates that if meticulous translation and back translation assessments are not completed prior to applying a validated scale from one linguistic culture into another, then data from the translated tool can be meaningless. It also shows that combining different linguistic reviewers (health experts, local residents, men and women) and methods (individual interviews, Focus Group Discussions, cognitive interviews) can lead to a consensus on the cultural equivalency of rewritten items.

An ethnographic study of a rural Australian farming community facing massive environmental degradation from coal mining documented significant expressions of distress due to loss of sense of place, well-being and control. Glenn Albrecht coined the term ‘solastalgia’ to describe the lived experience of loss and dislocation — “of being undermined by forces that destroy the potential for solace” derived from ones cherished environment (Connor, et al., 2004, p.55).

This construct capturing an Australian experience appears to be “transcultural,” stimulating global investigations. I have received permission requests to use this scale from over 50 master’s and PhD students, research supervisors, and international development agencies, from 20 countries so far. Requests have come from North and South America, Africa, the EU, South Asia, the Middle East, the Pacific, and other Australia universities.

Solastalgia has been applied to diverse causes of environmental degradation including bushfires, floods, deforestation, pollution, volcanic eruptions, mining damage and particularly the effects of climate change. It is associated with mental health outcomes (eco-anxiety/depression) either directly or as a mediating variable (Levison, et al., 2023); it relates to present as well as anticipatory loss (e.g, Stanley, 2023.) The present paper adds to this diverse literature by articulating the first crucial steps towards establishing a culturally valid measurement of solastalgia among Kenyans.

Specific Comments

1) Introduction

EDS. This Introduction gives a clear sense of how solastalgia is defined and used in the literature and its importance in understanding the emotional impact of environmental loss, particularly among minority and Indigenous populations, and those without resources to attenuate ecological loss from the climate crisis.

The authors describe the ‘full’ Environmental Distress Scale (EDS) as a multi-component model that contains items measuring perceptions of environmental change (hazards), threat appraisal, felt impact of changes as well as solastalgia and environmental action. However, as the report progresses, the authors focus exclusively on the solastalgia component of the EDS, and refer to it as the EDS. Given the extant literature on solastalgia, it would be appropriate to retain that label. Perhaps it could be labelled as the “EDS-Solastalgia”?

Solastalgia has taken on a life of its own in recent years, separate from the full EDS. It is evoked within the community generally, the arts, as well as the research literature where it is linked to a plethora of upsetting environmental impacts.

Transcultural. It is unclear how the authors define “transcultural translation.” In this instance, is it the same as “cross-cultural,” i.e., from English to Swahili? Transcultural implies use across multiple cultural settings. However, established procedures for ensuring cultural equivalency across the two languages is well described (e.g., comprehensive, relevance, completeness), as is the rationale for how they can be implemented.

2) Materials and methods

Further demographic and cultural information about the Kalifi County people would be useful to understand their relationship with the drought affected environment and the task of completing a questionnaire. Knowledge of leadership, religion, gender roles, education, literacy, and household economics as they relate to place attachment are important.

P6, line 30. Perhaps the word “inhospitable” rather than “insusceptible”

Participants. It would be clearer for the reader if Table 1 was presented earlier to define the 5 stages of engaging project participants. It is not a table of results per se. The table can be 5 x 5 (plus headings), and the first row is:

Stage 1: Translation of EDS-Solastalgia into Swahili and backtranslation (add relevant characteristics of the translator).

Stage 2: Expert Interviews (perhaps age range, but definitely range of years worked in these sub-counties).

Stage 3: Focus Group Discussions (add age range and average age by sex)

It should be made clear in the description that the Kaloleni FGD members were women and in Rabai they were men.

Stage 4: Consolidation of tool by two local lead researchers and local experts.

Stage 5: Cognitive interviews (add age range and length of time in area by sex)

Say more about how the cognitive interview participants were recruited. It seems they were bi-lingual and perhaps professionals? Were these one-on-one interviews? Were they tape recorded? Were they from the same organisations as the ‘experts’? Did the researchers have an existing relationship with these participants?

Describe how each stage unfolded in a separate paragraph, and how it related to subsequent stages (if it did).

Response set items judged by cognitive interviewees. A crucial piece of information is how participants understand and interpret the scale that is being used to respond to each statement. Was it the original 5-point scale using “strongly agree, agree, neither agree nor disagree, disagree, and strongly disagree”? It is stated the 10 cognitive participants were asked for reasons behind their use of the rating scale, and stated what experiences would be necessary for changes their responses up or down. These are deeply informative ‘think aloud’ techniques and details about their answers would be valuable in the results.

9-item Solastalgia questionnaire

It is appropriate that the authors have translated the full 9-item solastalgia scale that evolved from the original EDS validation study published in 2006 (Higginbotham, et al.) The wording and examples within the original scale were framed around degradation from open cut mines. This second-generation scale is more ‘generic.’ Validated through the longitudinal Hunter Community Study (tracking multiple health measures on older Australians), it asks respondents to focus on changes in their ‘local environment’, and the sense of distress that such changes may bring. Longitudinal study results found a pattern of associations with independent measures of psychological and biological distress (Higginbotham, et al., 2007).

Recently, researchers undertaking large scale surveys on bushfires and climate change have sought to validate a brief (and “economical”) 5-item solastalgia scale (Christenson, et al., 2024; Levison, et al., 2023; Stanley et al., 2024). Doing so may help address the issue of researchers (who require brief versions)choosing items based on subjective criteria.

However, in the context of the Swahili translation, I believe it is more valuable to be mindful of the breadth (content) of the original construct, given that brief versions may lose elements that are highly pertinent to the Kenyan culture groups facing drought. The published 5-item brief scale reduced several original components including: undermining/loss of sense of place; imposed transformation/powerlessness; and solace derived from one’s environment is undermined.

Results.

Qualitative methods are valuable for revealing cultural understandings, meanings and lived experiences of people encountering environmental change in their place. Indeed, qualitative studies are most compelling when they tell a story about a people and their place. Results here are described by method and reveal how words from the initial translation could be rewritten to improve their fit and interpretability among the local Swahili residents. The explanations for new word choices are clear and persuasive. Clearly, getting different groups of people to interrogate the meaning of scale questions in different ways clarifies what is being asked and whether it captures the intended meaning.

However, when the results are described by method (e.g., expert interview, FGD, cognitive interview), it leads to repetition when the same word use issue is reported across methods, such as replacement of the word ‘aspects’ with the simple word, ‘things.’ The authors might consider refocusing the results section to present the most significant word changes suggested across the methods, and then explain those changes from a cultural, linguistic and environmental point of view. In other words, are there patterns (themes) in why a word choice is a better fit. The first two paragraphs of the discussion accomplish this to some extent.

Table 2 is somewhat unwieldy. Useful as an appendix. The items should be numbered in Table 3. Item 7, “Unique aspects…,” uses the word unique twice, which is awkward in English. Item 2 uses ‘getting depleted/lost’ while item 3 uses “being lost/depleted.” The word “being” sounds better to me and consistent word order is useful.

Discussion.

Have the qualitative methods suggested any environmental losses felt by these drought-stricken communities that could add to the understanding of solastalgia as a concept? Are there local components/dimensions of distress from such ‘unusual’ changes that are missing (i.e., culture specific idioms of distress)? The study did not set out to do this, but it might be something to bring up in the discussion.

It was surprising that the fine-grained approach of the cognitive interviews did not add further to the translation, or perhaps the broader questions just mentioned. Is there anything about the context of the cognitive interviews that favoured agreement with what was already written (e.g., wishing to appear polite)? What were their instructions for this task?

References

Christensen, B.K., Monaghan, C., Stanley, S.K. et al. (2024). The Brief Solastalgia Scale: A Psychometric Evaluation and Revision. Ecohealth, 21(1): 83–93.

Connor, L., Albrecht, G., Higginbotham, N. et al. (2004). Environmental change and human health in Upper Hunter communities in New South Wales. EcoHealth 1 (Supp 2): 47-58.

Higginbotham, N., Connor, L., Albrecht, G., et al. (2006). Validation of an Environmental Distress Scale. EcoHealth, 3(4), 245-254.

Higginbotham, N., Bowe, S., M. McEvoy, M., Freeman, S., Attia, J., Albrecht, G. (2007). Clinical validation of ‘solastalgia’: A component of environmental distress. The Asia Pacific Eco Health Conference, Melbourne, Victoria. 30 November – 3 December 2007.

Levison, Z., Stanley, S.K., Rodney, R.M. (2023). Solastalgia mediates between bushfire impact and mental health outcomes: A study of Australia’s 2019-2020 bushfire season.

Journal of Environmental Psychology, 90 (pages 1-11).

Stanley, S. (2023). Anticipatory solastalgia in the Anthropocene: Climate change as a source of future-oriented distress about environmental change. Journal of Environmental Psychology, 91(sup1):102134.

Stanley, S., Heffernan, T., Macleod, E. (2024). Solastalgia following the Australian summer of bushfires: Qualitative and quantitative insights about environmental distress and recovery. Journal of Environmental Psychology, 95(sup1):102273

---

## [Editor Report]

Thank you for submitting the revised manuscript. Given the reviewers recommendations can you kindly address the following issues in particular:

1. Ensuring that all methodological aspects of the study are addressed. Were appropriate, pinpoint the potential limitations of the applied methods and perhaps provide recommendations for future studies.

2. The conclusions must be in keeping with study’s strengths and weaknesses 

3. The use of international guidelines to guide the reporting of the methodology is highly encouraged. For instance, the reders could refer to the COSMIN guidelines for cross-cultural adapation and validation studies - https://www.cosmin.nl/

4. Given the variation in validation studies terminology, the authours must explicitly declare the type of pyschometric properties that were evaluated; consideration should be made towards possible title rephraising. 

The reviewers’comments are attached for your consideration.

---

## [Reviewer Report]

The manuscript “Cultural and contextual adaptation of the Environmental Distress Scale in Kilifi, Kenya, affected by drought: A cross-cultural translation and cultural adaptation”, is the review of the presented process of translation and cultural adaptation of a scale for the assessment of aspects related to environmental perception. The theme is current, relevant and important in the environmental and geopolitical context not only of the African continent, but internationally. The technical language aims to clearly discuss the aspects that justify the need to use a scale to assess the environmental effects of community residents in Kenya. The present review sought to contribute to the qualification of the manuscript and the possibility of disseminating the research to the international community, in view of the current importance of the theme at the end of COP 29 and preparations for COP 30.

It was suggested that the authors rethink the issue of the title to avoid the circularity of terms, and a request was granted.

The unclear expressions (translation and rigorous cross-cultural adaptation) as well as the rigorous expression do not proceed to what is suggested and have been suppressed throughout the manuscript.

The impact statement, line 43 was met.

In the introduction, line 19, the aspects were taken into account.

In the 25-line method, participants, the authors characterize both the translation, and the back-translation was rectified.

The tables referring to the composition of the participants are presented and were adequate. Table 2 was adequate. It is requested that no tables be presented in the discussion topic, only in the results.

The conclusions are presented succinctly.

---

## [Reviewer Report]

It is valuable to learn that this cultural adaptation of Solastalgia is part of a larger project that includes an ethnographic study, and a psychometric validation of the Kenya Solastalgia scale. I would encourage the authors to write and publish a report that combines all three of these studies, to give the richest possible understanding of Solastalgia in this environment, and to provide a model for other researchers showing the vital synthesis of ethnographic, translational and psychometric sources of knowledge to provide the deepest understanding of this construct.

---

## [Reviewer Report]

The manuscript “Cultural and contextual adaptation of the Environmental Distress Scale in Kilifi, Kenya, affected by drought: A cross-cultural translation and cultural adaptation”, is the second revision. It is characterized by the topicality of the theme and its relevance and also important in the environmental and geopolitical context of the African continent, as well as internationally. The technical language aims to clearly discuss the aspects that justify the need to use a scale to assess the environmental effects of community residents in Kenya. The present version sought to contribute to the qualification of the manuscript and the authors made adjustments requested in the previous review. The rectification of tables shows compliance and adequacy, and one on the main sociodemographic characteristics is included, which facilitates the reader’s understanding.

The reviewer notes that he evaluated the second manuscript presented in the download file (starting on page 23 of the pdf) and without revision marks (which appear in the first manuscript of the document).